# Structure and mechanism of the RalGAP tumor suppressor complex

René Rasche [1,6], Björn Udo Klink [2,3,6], Lisa Helene Apken [4], Esther Michalke[4], Minghao Chen [2,3], Andrea Oeckinghaus[4,5], Christos Gatsogiannis [2,3] ✉ & Daniel Kümmel [1] ✉

The RalGAP (GTPase activating protein) complexes are negative regulators of the Ral GTPases and thus crucial components that counteract oncogenic Ras signaling. However, no structural information on the architecture of this tumor suppressor complex is available hampering a mechanistic understanding of its functionality. Here, we present a cryo-EM structure of RalGAP that reveals an extended 58 nm tetrameric architecture comprising two heterodimers of the RalGAPα and RalGAPβ subunits. We show that the catalytic domain of RalGAPα requires stabilization by a unique domain of RalGAPβ, providing the molecular basis for why RalGAP complexes are obligatory heterodimers. Formation of RalGAP tetramers is not required for activity in vitro, but essential for function of the complex in vivo. Structural analysis of RalGAP subunit variants reported in cancer patients suggests effects on complex formation and thus functional relevance, emphasizing the significance of the obtained structural information for medical research.

The Ral GTPases RalA and RalB act as molecular switches that regulate vesicular trafficking and signal transduction, thereby impacting exocytosis, endocytosis, cell survival, and proliferation[1]. The best studied Ral effectors are the exocyst subunits Sec5 and Exo84[2–4] and the Ral binding protein 1 (RalBP1, or RLIP76)[5–7], which all three provide a link to membrane transport. Via Sec5, RalB has also been shown to activate tank-binding kinase 1 (TBK1) in an exocyst-independent manner to promote tumor cell survival[8]. Importantly, the Ral signaling pathway has long been recognized for its crucial function downstream of the Ras GTPases[9]. Rals have been shown to support Ras-dependent tumorigenesis but have also been implicated in cancer progression independent of Ras[1].

The RalA and RalB isoforms are 82% identical and belong to the Ras superfamily of small GTPases. The activation of Rals from a GDP-bound "off" state to a GTP-bound "on" state is mediated by guanine nucleotide exchange factors (GEFs), some of which are directly controlled by the Ras GTPases[10]. The two RalGAP complexes, RalGAP1 and

RalGAP2, act as GAPs (GTPase activating proteins) that limit Ral-GTP levels by promoting hydrolytic activity. They comprise one of the catalytic α-subunits RalGAPα1 (RGα1) in the RalGAP1 complex or RalGAPα2 (RGα2) in the RalGAP2 complex, and a common regulatory β-subunit (RGβ)[11–13] (Fig. 1a). Both RGα and RGβ contain asparagine thumb GAP(-like) domains. First identified in RapGAP, this type of GAP domain uses a unique catalytic mechanism by employing a conserved asparagine to position a water molecule in the GTPase nucleotide-binding pocket to promote GTP hydrolysis[14–16]. The GAP domain of RGβ lacks the catalytic asparagine residue and is thus not active[11], but RalGAP complexes require the conserved asparagine of the GAP domain of the RGα subunits for activity. However, RGα alone is not functional and depends on the binding of RGβ with the underlying molecular details remaining unclear[11].

Another RapGAP domain-containing protein is tuberous sclerosis complex 2 (TSC2), which together with TSC1 and TBC1D7 forms the TSC complex[17–19]. The TSC2 GAP domain accelerates GTP hydrolysis of

[1]Institute of Biochemistry, University of Münster, Münster, Germany. [2]Institute for Medical Physics and Biophysics, University of Münster, Münster, Germany. [3]Center for Soft Nanoscience (SoN), University of Münster, Münster, Germany. [4]Institute of Molecular Tumor Biology, University of Münster, Münster, Germany. [5]Department Metabolism, Senescence and Autophagy, Research Center One Health Ruhr, University Alliance Ruhr & University Hospital Essen, Essen, Germany. [6]These authors contributed equally: René Rasche, Björn Udo Klink. ✉e-mail: christos.gatsogiannis@uni-muenster.de; daniel.kuemmel@uni-muenster.de

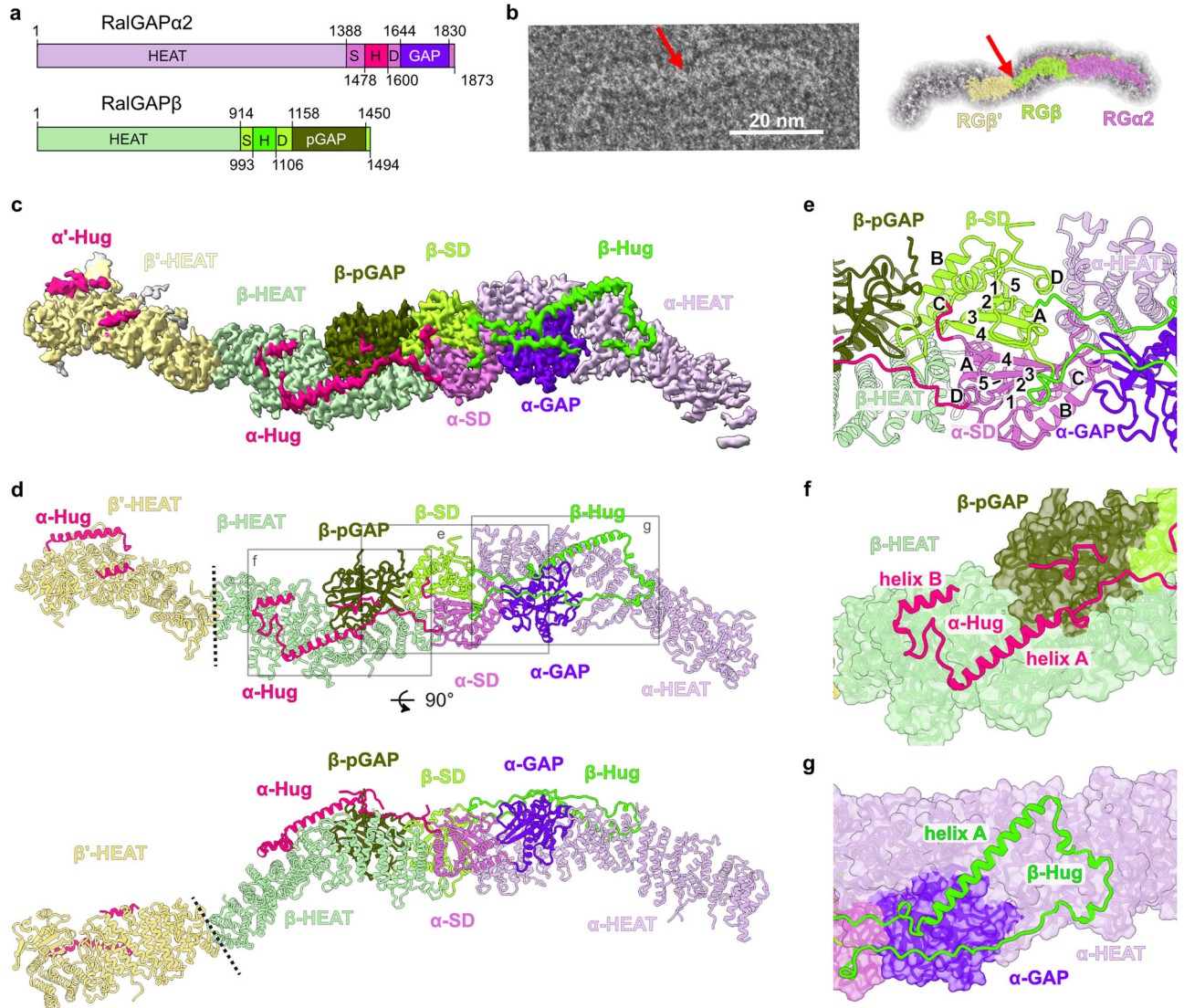

**Fig. 1 | Cryo-EM structure of the RalGAP complex. a** Domain architecture of RGα2 and RGβ. HEAT: α-solenoid HEAT repeat-like domain; SD split stabilization and dimerization domain; H hug domain; (p)GAP (pseudo-) GTPase activating protein domain. **b** Negative stain image of a full RalGAP particle in comparison to the 3D reconstruction of the half-particle. The red arrow marks the connection between two heterodimeric half-particles. **c** Composite experimental map of RalGAP (half)-

particles. **d** Model of the RGα2/RGβ heterodimer and the N-terminal portion of a second RGβ subunit. Domains are colored as in 1 A. Dashed lines indicate the interface of both half-particles. **e** Close-up of the SD domain heterodimerization interface. The β-strands (1–5) and α-helices (A–D) of the SD domains are labeled. **f** Interaction of the RGα2 hug domain with RGβ. **g** Interaction of the RGβ hug domain with RGα2.

the GTPase Rheb[16,20–23]. Because Rheb is required for the activation of the mTORC1 (mechanistic target of rapamycin complex 1) kinase[24], the master regulator of cellular growth, the TSC complex is an important tumor suppressor and loss of function causes the genetic disease TSC[25].

In addition, the non-canonical κB-Ras GTPases (of which two isoforms, κB-Ras 1 and κB-Ras 2 are present in humans) bind to the N-terminal region of both RGα subunits and support complex function in vivo with the underlying molecular reason remaining elusive[26,27]. The activity of RalGAP can furthermore be negatively regulated by AKT phosphorylation and 14-3-3 protein binding[12,13,28], which is also reminiscent of findings reported for the TSC complex[29].

Despite the ability of Ral GTPases to promote tumorigenesis, no Ral mutations with oncogenic potential analogous to the Ras GTPases have been described in patients, and neither has relevance of mutations in Ral effectors or regulators been studied[30]. However, low expression of RalGAPα2 leads to an increase in tumor cell proliferation, invasion, and migration in cancer cell lines or mouse models of

hepatocellular carcinoma[31], colitis-associated cancer[32], oral squamous cell carcinoma[33], prostate cancer[34], and bladder cancer[35]. A comprehensive proteomic analysis of 140 patient pancreatic tumor samples also has revealed significantly reduced RalGAPα2 expression levels in combination with strong increase of inhibitory phosphorylation[36]. Knockout of RalGAPβ increases migration and invasion of PDAC (pancreatic ductal adenocarcinoma) cell lines as well as xenograft growth in vivo and accelerates local tumor growth and metastasis[37]. Furthermore, κB-Ras 1/2 deficiency in a KRas^{G12D}-driven PDAC mouse model caused faster development of invasive carcinoma and shorter life span of animals[27]. In patients, lower RalGAPα2 expression levels correlate with an invasive phenotype in colitis-associated cancer[32]. They also inversely correlate with survival in prostate[38] and bladder cancer[35], and κB-Ras expression levels are reduced in human PDAC tumor samples[26]. These data indicate that RalGAP complexes are important tumor suppressors.

Interestingly, the bi-allelic inactivation of *RALGAPA1* was reported to cause a rare genetic disease leading to impaired neurodevelopment,

infantile spasms, muscular hypotonia, and feeding abnormalities[39], suggesting additional medical relevance of RalGAP beyond cancer.

Here we provide a structural and biochemical characterization of the RalGAP complex and thus a solid framework for understanding the mechanisms underlying its function. RalGAP assembles into a dimer of heterodimers, which results in the formation of an extended 58 nm structure. We demonstrate that this complex architecture is essential for functionality and explain why RGα2 requires RGβ binding to maintain an active conformation. Based on this detailed structural framework, we analyze RalGAP patient variants, suggesting a possible role in pathological processes such as tumorigenesis.

## Results

### Cryo-EM structure of the RalGAP complex

We established expression of the RGα2/RGβ/κB-Ras2 complex in Expi293F cells and purified it via FLAG affinity purification and size exclusion chromatography to obtain a monodisperse sample for cryoEM single particle studies (Supplementary Fig. 1a, b). The sample shows a homogenous set of characteristic extraordinarily long particles that resemble the shape of a mustache (Supplementary Fig. 1c). Already at the initial steps of image processing, it became evident that the 58 nm elongated particles consist of two identical subcomplexes, which are, however, flexibly linked to each other. Because of this internal motion, the 2D class averages obtained of the complete Ral-GAP did not allow the generation of a 3D model (Supplementary Fig. 1d). Therefore, we focused our analysis on the half-particle, which represents the unique portion of the "mustache" structure (Supplementary Fig. 2, Supplementary Table 1). In addition, this focus was helpful for efficient processing, as we required a box size of ~42 nm for the half-particle alone. This yielded a map with an overall resolution of 3.8 Å. To achieve better local resolution, we performed multi body refinement for the core as well as the terminal portions of the "half-mustache". The resolution of the focused maps was 4.4 Å at the hinge region between the two "half-mustaches", 3.8 Å for the central region of the RGα2/RGβ heterodimer, and 3.9 Å for the peripheral region at the tip of the complex (Fig. 1c, Supplementary Figs. 2 and 3). We were able to unambiguously build an atomic model of the RGα2/RGβ heterodimer using AlphaFold2 models as a starting point (Fig. 1d). Several loop regions in both subunits, the very N-terminal portion of RGα2 (residues 1-73) and κB-Ras2, which binds to this region[26], were not resolved and could not be modeled. However, the reconstruction includes a portion of a second RGβ subunit of the adjacent "half-mustache", which allowed us to unambiguously model the N-terminal portion of the second RGβ subunit and a small part of the second RGα2 subunit (Fig. 1c, d).

Both RGα2 and RGβ have an overall similar architecture (Fig. 1a, Supplementary Fig. 4a,b), including an N-terminal α-solenoid HEAT repeat-like domain, with frequent insertions of loops and additional helices within and between HEAT motifs. This region is followed by a globular domain (referred to as stabilization and dimerization (SD) domain in the following) and a GAP domain that is inserted between two helices $\alpha^B$ and $\alpha^C$ of the SD domain (Fig. 1e).

The SD domains are the central organizing module of the RalGAP complex and mediate heterodimerization of RGα2 and RGβ in a tail-to-tail fashion, resulting in an overall arch-like shape of the RalGAP heterodimer. In both RalGAP subunits, they show a $\beta^1$-$\beta^2$-$\beta^3$-$\beta^4$-$\alpha^A$-$\beta^5$-$\alpha^B$-$\alpha^C$-$\alpha^D$ topology, forming a central 5-stranded β-sheet with two helices on each side (Fig. 1e). The β-strands from both SD domains form one continuous sheet to mediate heterodimerization (Fig. 1e). In addition, the SD domains interact in trans with the C-terminal HEAT repeats of their binding partner. Interestingly, between β4 and αA of both RGα2 and RGβ SD domains, large loop regions with little secondary structure except for two helical segments are inserted. These loops reach over from RGα2 to RGβ and from RGβ to RGα2, respectively, thereby linking the subunits with each other and accounting for one-third of the

heterodimer interface (buried surface RGα2-RGβ: 10,443 Å², RGα2-RGβ$^{hug}$: 3923 Å², RGα2$^{hug}$-RGβ: 3574 Å²) (Fig. 1f, g). We therefore termed these structural motifs "hug" domains and "hug" helices.

The general domain organization of RGα and RGβ is reminiscent of the TSC2 subunit of the TSC protein complex (Supplementary Fig. 4c), suggesting that RGα, RGβ and TSC2 are likely derived from a common ancestor. However, while TSC2 forms a homodimer via its SD domain, RGα2 and RGβ form a heterodimer. Because the twists of the solenoid domains are different, the overall shapes of RGα2/β and TSC2 dimers are distinct (Supplementary Fig. 4d). There are also differences in the structure of the RalGAP and TSC2 SD domains. The SD domain of TSC2 forms homodimers via a shorter β-sheet and has multiple additional helices inserted between the β-strands, which together form a domain termed "saddle"[23] that stabilizes binding of the TSC1 subunit. Instead of a "saddle" domain, RalGAP subunits carry the "hug" domain insertion. These distinct features of the SD domains are thus required to realize the different complex assemblies and architectures of the TSC and RalGAP complexes.

### Comparison of a Asn-thumb GAP domain and a pseudo-GAP domain in RalGAP

The C-terminal domain of RGα2 is reported to confer the GAP activity towards Ral. It folds with a $\beta^1$-$\alpha^A$-$\alpha^B$-$\beta^2$-$\beta^3$-$\beta^4$-$\alpha^C$-$\beta^5$-$\beta^6$-$\beta^7$-$\beta^8$-$\alpha^D$ topology that resembles the Asn-thumb GAP domains of TSC2 and RapGAP (Supplementary Fig. 4e–g), with a characteristic catalytic asparagine at position 1742 at the end of helix $\alpha^C$ (Fig. 2a). RGβ has a similarly folded domain with the core β-sheet structure, but large loops are inserted between $\beta^2$-$\beta^3$ and $\beta^5$-$\beta^6$. Importantly, the catalytic $\alpha^C$ helix is substituted by an extended loop (aa 1264-1341) that is partially unresolved in the EM density (aa 1272-1329), rendering the domain a pseudo-GAP (pGAP) lacking the active site residues (Fig. 2b).

For a better understanding of the GAP activity mechanism, we modeled the interaction of RalA with the full RalGAP complex. We first generated an AlphaFold3 prediction[40] of RalGAPα2/β in complex with RalA and GTP and positioned Ral on the experimental structure by superposing the RGα2 GAP domains of model and prediction (Supplementary Fig. 5a–c, Fig. 2c). The GAP domain of the predicted Ral/RalGAP2 model superimposes well with the GAP domain of the homologous experimental Rap1A/Rap1GAP structure (PDB: 3BRW[15]) with an RMSD of 1.128 Å over 115 Cα atoms (Supplementary Fig. 5d, e). The catalytic asparagine residue (N1742) is properly positioned 4.4 Å away from the γ-phosphate to coordinate a nucleophilic water molecule for stimulation of GTP hydrolysis (Fig. 2d). Rap and Rheb have a conserved tyrosine residue in their switch I region that was shown to support GAP-stimulated GTP hydrolysis by stabilizing the γ-phosphate of GTP[15,16,41]. Tyrosine 43 of RalA is modeled to occupy a similar position (Fig. 2d) and thus likely fulfills the same function as in the homologous GTPases. In addition to the GAP domain, the SD domain of RGα2 is predicted to make contacts with the GTPase (Fig. 2c) similar to how the TSC complex is proposed to bind Rheb[23]. This interface likely contributes to affinity of the interaction and specificity of GTPase recognition by their cognate GAPs.

### The unique hug domain of RGβ contributes to catalytic activity through stabilizing effects

Surprisingly, while RGβ is essential for GAP activity in cells and in vitro[11], it does not interact with RGα2 at the active site and - based on the high confidence AlphaFold3 model and in homology to RapGAP - is also unlikely to interact with Ral (Fig. 2c). However, the RGβ hug domain makes extensive contacts with RGα2, in particular the long hug helix that runs alongside the HEAT and GAP domains and holds them together (Fig. 1g). We suspected that the β-hug domain thus may act as scaffolding domain and be indirectly involved in GAP activity. To first test the relevance of the hug domains for complex stability, we generated deletion constructs RGα2$^{Δhug}$ (lacking aa 1498-1593) and

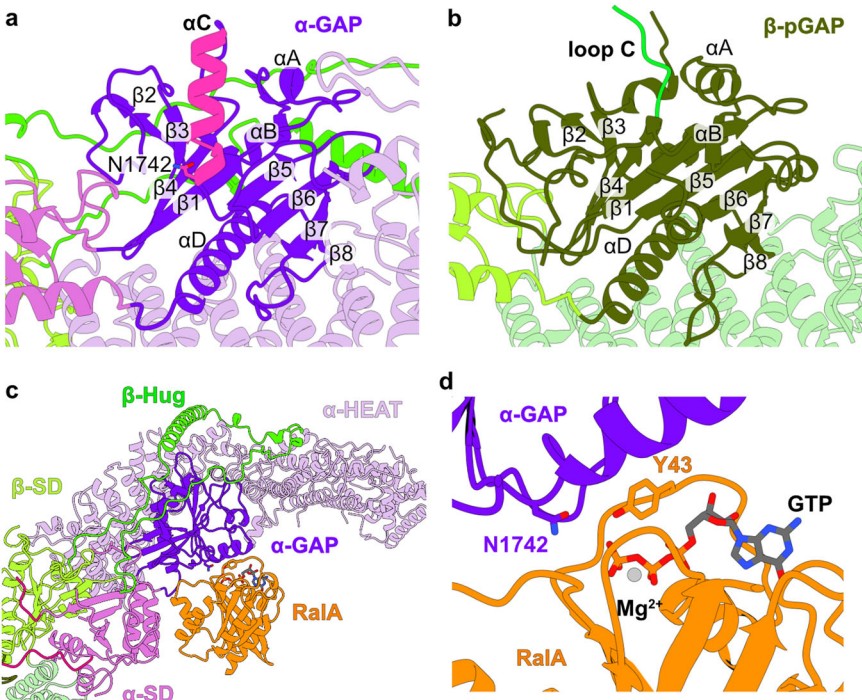

**Fig. 2 | GAP domains and Ral binding of RalGAP. a** Experimental structures of the Asn-Thumb GAP domain of RGα2 and **b** the pseudo-GAP domain of RGβ in the same orientation. **c** Model of GTP-bound RalA associated with RGα2 based on an AlphaFold 3 prediction. **d** Close-up of the active site. The catalytically relevant amino acids RGα2$^{N1742}$ and Ral$^{Y43}$ are positioned like the equivalent residues in the structure of the Rap-RapGAP complex.

RGβ$^{Δhug}$ (lacking aa 1004-1099) lacking these domains. In co-immunoprecipitation experiments, RGα2$^{Δhug}$ still bound full-length RGβ, while RGβ$^{Δhug}$ almost completely lost its ability to associate with RGα2 (Fig. 3a), revealing an essential role of RGβ$^{hug}$ in complex assembly. Expectedly, deletion of the hug domain in both subunits abolished complex formation as well. Furthermore, we found that the RGβ hug domain alone was sufficient to form a complex with RGα2 (Fig. 3b). These findings suggest that the hug domain of RGβ is crucial for the integrity of the RalGAP complex.

For functional validation, we analyzed the importance of the RGβ hug domain with a cellular assay where RalA and RGα2 were co-expressed with the different RGβ constructs. The levels of active RalA in these cells were determined by a pull-down assay with the Sec5 Ral effector domain that selectively binds GTP-bound Ral GTPases. RGα2 and RGβ together were able to sustain RalGAP activity, but not the co-expression of RGα2 with RGβ$^{Δhug}$ (Fig. 3c). We then used RGβ knock-out mouse embryonic fibroblasts (MEFs), which showed dramatically increased levels of active Ral, for reconstitution experiments. Rein-troduction of full-length RGβ, but not RGβ$^{Δhug}$, was able to reduce Ral activity in these cells (Fig. 3d). To identify a minimal active RalGAP complex, we employed an HPLC-based in vitro GAP assay that mea-sures GTP hydrolysis by RalA in the presence of RalGAP (Fig. 3e). As previously reported, robust stimulation of GTP hydrolysis was observed for the full-length RGα2/RGβ complex, but not for RGα2 alone[11]. We then co-expressed and purified RGα2 with fragments of RGβ comprising only the hug domain (RGβ$^{hug}$, aa 973-1118), the C-terminal SD domain with hug and pseudo-GAP domain (RGβ$^{SD-hug-pGAP}$, aa 881-1495) and a slightly longer construct that additionally contains the C-terminal portion of the HEAT repeat domain (RGβ$^{cHEAT-SD-hug-pGAP}$, aa 576-1494). Although these constructs were less stable than full-length RGβ, they could be co-purified with RGα2 and these sub-complexes caused robust acceleration of GTP hydrolysis by RalA (Fig. 3e). This identifies the hug domain of RGβ as the key element sufficient to render RGα2 catalytically active. Taken together, the structural and biochemical data suggest that the function of RGβ in the enzymatic mechanism of the RalGAP complexes is to stabilize the GAP domain via its hug domain.

## Tetramerization of RalGAP

Unexpectedly, the structure reveals homodimerization of RGβ via its N-terminal region at the central interface between the two half-particles (Fig. 1b). Thus, RGβ mediates the interaction of two RGα2/RGβ heterodimers and leads to the formation of the tetrameric 'mus-tache'-shaped RalGAP complex. The RGβ subunits interact via hydro-phobic residues in the first two helices of the RGβ HEAT domain (Fig. 4a, b). The key amino acids of this interface include a central tryptophane (W65) and a series of valines in the first helix (V29, V33, V37). The small portion of the second RGβ' subunit visible in the experimental map (Fig. 1c) provides sufficient detail to extend the model of RalGAP by superposition of a RGα2/RGβ heterodimer with the RGβ' N-terminal domain and experimental images of the full par-ticle. This allowed us to create a complete model of the extended, ~580 Å long RalGAP tetramer (Fig. 4c, d).

## Formation of RalGAP tetramers via RGβ dimerization is required for complex function

We next asked if homodimerization via the N-terminal domain of RGβ is functionally relevant. In co-immunoprecipitations of differently tagged versions of RGβ we observed that indeed, two RGβ copies can interact in cells (Fig. 5a). We then introduced mutations of key residues identified in the observed homodimer interface of RGβ (Fig. 4a, b). A single point mutant RGβ$^{D1}$ (W65R), a triple dimer interface mutant RGβ$^{D3}$ (V29E, V33Q, V37E), and a combination of both RGβ$^{D4}$ all lost the ability to homodimerize (Fig. 5a), while interaction with RGα2 was not affected (Supplementary Fig. 6a). Co-purification of RGβ$^{D4}$ with RGα2 yielded smaller complexes compared to wildtype as assessed by size exclusion chromatography (Supplementary Fig. 6b) and negative stain EM analysis (median 577 Å and 296 Å, respectively, Fig. 5b, Supple-mentary Fig. 6c, d). RGα2/RGβ$^{D4}$ complexes showed stimulation of GTP hydrolysis in vitro comparable to wild-type RalGAP, demonstrating

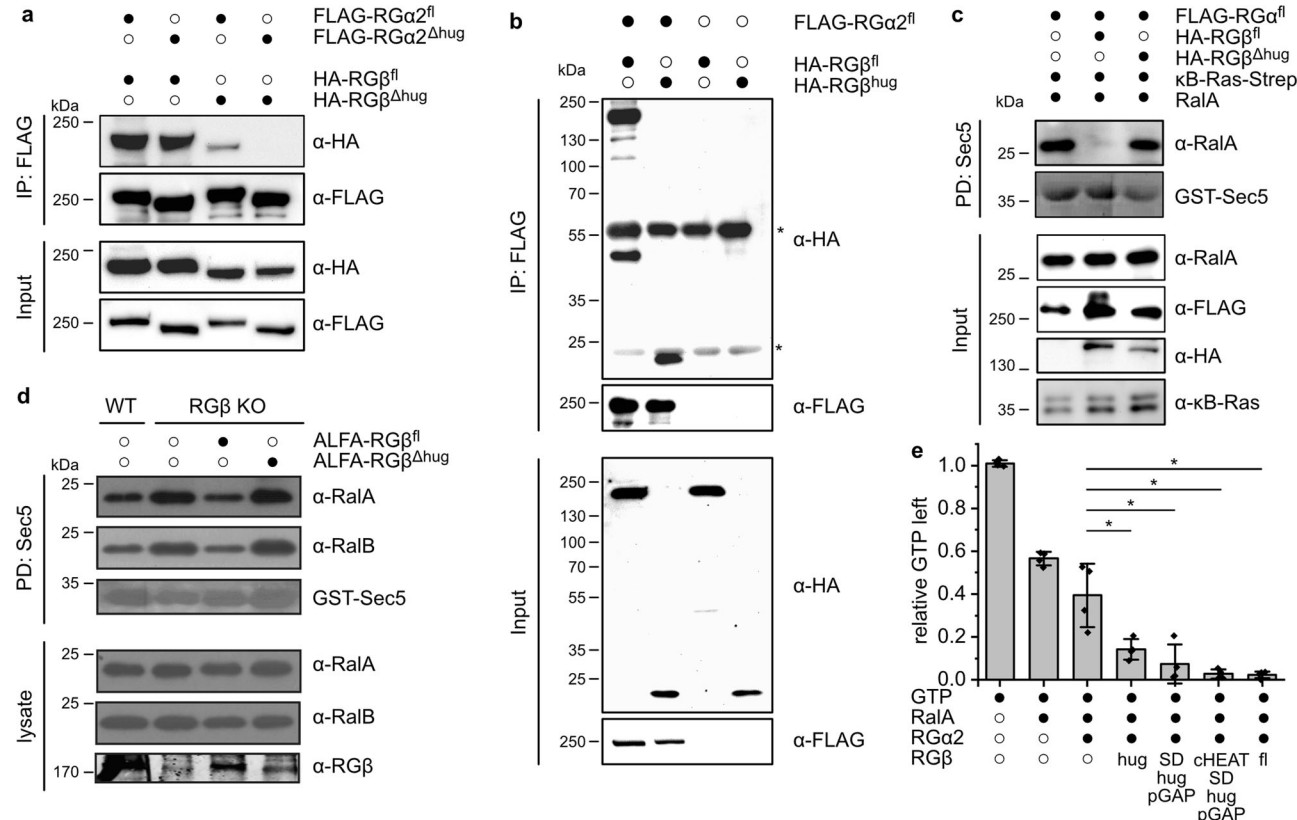

**Fig. 3 | The RGβ hug domain is required for RGα2 GAP activity. a** Co-immunoprecipitations of RGα2 and RGβ full-length (RGα2$^{fl}$, RGβ$^{fl}$) and hug domain deletion constructs (RGα2$^{Δhug}$, RGβ$^{Δhug}$) from transiently transfected HEK293T cells. **b** Co-immunoprecipitations of RGα2 with RGβ full-length and a construct only comprising the β-hug domain (RGβ$^{hug}$, aa 973-1118) from transiently transfected HEK293T cells. **c** Sec5 pull-down from HEK293T cells transfected with RalA and RGα2 without or with RGβ$^{fl}$ and RGβ$^{Δhug}$. **d** Sec5 pull-down analysis of RGβ knock-out MEFs reconstituted with RGβ$^{fl}$ or RGβ$^{Δhug}$. **e** In vitro GAP assay of RalA GTP

hydrolysis by RGα2 without or with various RGβ truncation constructs. The same molarity of RGα2 was used in each assay and the activity was normalized to the amount of RGβ present in the individual reactions, n = 4 independent experiments, two-tailed t-test with Welch-correction, *p = 0.0358 (RGα2- RGα2/RGβ$^{hug}$); *p = 0.0140 (RGα2-RGα2/RGβ$^{SD-hug-pGAP}$); *p = 0.0146 (RGα2-RGα2/RGβ$^{cHEAT-SD-hug-pGAP}$:); *p = 0.0146 (RGα2-RGα2/RGβ$^{fl}$); data are represented as means ± standard deviation.

that catalytic activity is not affected by the mutations (Fig. 5c). Surprisingly, RGβ$^{D4}$ expression in RGβ KO cells was not able to rescue the phenotype but resulted in unchanged high levels of GTP-bound Ral (Fig. 5d). We cannot exclude secondary effects of the mutations in vivo. However, our data strongly suggest that an intact tetrameric RalGAP assembly is required for its physiological function, possibly by ensuring proper regulation or localization.

**Structural analysis suggests pathogenicity of RalGAP variants identified in cancer patients**

To provide structural insights into the role of RalGAP complexes as tumor suppressors, we searched for missense mutations in the *RALGAPA1*, *RALGAPA2*, and *RALGAPB* genes in cancer patients. RGα1 and RGα2 are ~45% identical, suggesting that their structure and interaction with RGβ are conserved. We thus created an AlphaFold3 model of an RGα1/RGβ complex (Supplementary Fig. 7a, b), which indeed closely resembles the experimental structure of RGα2/RGβ (Supplementary Fig. 7c). The predicted domain architecture of RGα1, comprising an N-terminal α-solenoid, SD domain with "hug" domain insertion and a C-terminal GAP domain, corresponds to that of RGα2 (Supplementary Fig. 7d). The RGα1 hug domain is larger and the α-solenoid is modeled in a more bent configuration compared to RGα2, which would require experimental confirmation. However, from our characterization of RalGAP there is no indication that these structural differences would have a functional impact. Importantly, α-hug and β-hug interactions in RGα1/RGβ are modeled similar to what was observed

for RGα2/RGβ. We therefore considered mutations in RGα1 and RGα2 analogous for our variant analysis.

In the TCGA PanCancer Atlas Studies (32 studies, 10967 samples), RalGAP coding genes showed the highest missense mutation frequency in uterine and skin cancers (16% of all samples) (Supplementary Fig. 8a). We compiled a list of missense mutations in *RALGAP* genes from uterine and skin cancer patients reported on cBioportal[42,43], which are classified as variants of uncertain clinical significance (VUS). Their locations are spread over the entire length of the three genes and no hot spot mutations can be identified (Supplementary Fig. 8b–d).

We chose all positions that are reported mutated more than once in the *RALGAPA2* and *RALGAPB* genes for closer inspection. Comparison to the AlphaMissense database[44] yielded only a small portion of variants to be categorized as likely pathogenic (20%, Supplementary Table 2). AlphaMissense primarily classifies variants as likely pathogenic that are expected to directly affect protein folding and/or stability based on their predicted structure. Manual inspection of these variants on the basis of the experimental model confirms the assessment that the mutations are likely to negatively affect protein structure (Fig. 6a). However, AlphaMissense analyses proteins in isolation, but a large portion of variants classified as likely benign by AlphaMissense maps to protein-protein interfaces that are revealed by our complex structure (Fig. 6a). This includes the Ral binding site, the heterodimerization interface, and the hug domains or hug binding interfaces. Distortions of these interfaces are likely to interfere with

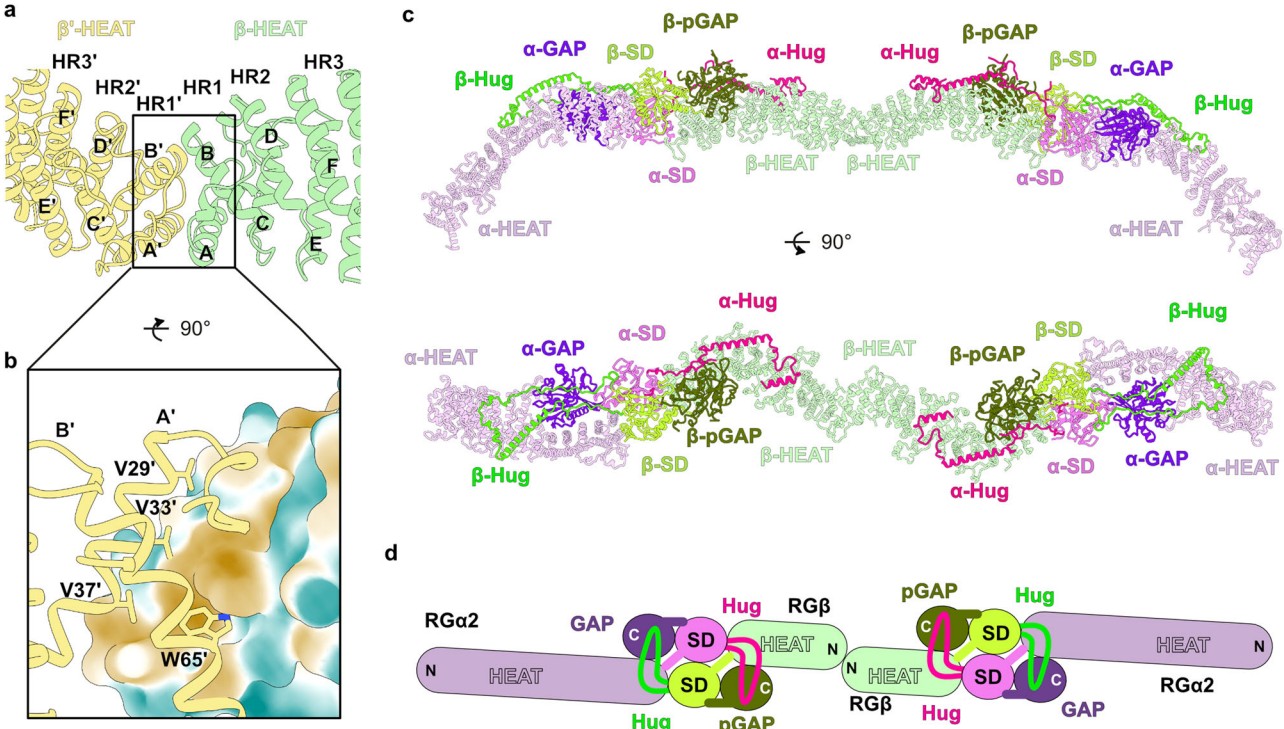

**Fig. 4 | Tetramerization of RalGAP. a** N-terminal homodimerization of RGβ. **b** Close-up of the homodimerization interface of the RGβ N-terminal domain. One RGβ subunit is shown in surface representation colored according to polarity (orange: hydrophobic, cyan: hydrophilic), the other RGβ is shown as cartoon with key interface residues shown as sticks and labeled. **c** Model of the RalGAP tetramer obtained by superposition of a second RGα2/RGβ heterodimer on the N-terminal HEAT domain of RGβ' in the experimental model. **d** Schematic representation of the RalGAP assembly.

complex function, could render RalGAP inactive, and thus the reported variants may potentially be pathogenic.

One example is the RGα2 variant R1738H, which maps to the catalytic helix of the GAP domain and is predicted to interact with RalA E97 via a salt bridge (Fig. 6b). Loss of this interaction is likely to interfere with the GAP activity of RalGAP. For further analysis, we investigated variants reported in cancer patients that map to the two essential interfaces - β-hug binding to RGα2 and RGβ N-terminal homodimerization - we validated by biochemical and cell biological studies. Variants are frequently found in both the β-hug domain and the corresponding binding interface on RGα2 (Fig. 6c). Furthermore, a cluster of patient variants is found at the RGβ homodimerization interface (Fig. 6d). Thus, the analysis of patient variants in the light of the RalGAP structure and its functional characterization suggests that at least some of the reported variants may be pathogenic and could contribute to tumorigenesis. Interestingly, a few variants cluster in a large disordered loop of the RGα2 HEAT domain (672-911), close to the reported AKT phosphorylation sites S696 and T715[28] (Supplementary Fig. 8, Supplementary Table 2). These variants may therefore be pathogenic without affecting RalGAP structure.

Independent of somatic mutations, bi-allelic loss of *RALGAPA1* was described as the cause for the genetic disease NEDHRIT (neurodevelopmental disorder with hypotonia, neonatal respiratory insufficiency, and thermodysregulation)[39]. Interestingly, one patient carried an allele with a pathogenic missense mutation (c.3227 A > G, p.Asn1076Ser). This position is conserved between RGα1 and RGα2, and the corresponding residues (N1076 of RGα1 and N943 of RGα2) are involved in stabilizing the HEAT repeat domain at the β-hug binding site of the protein (Fig. 5e). Mutation of RGα2 N943 (and likely RGα1 N1076) is thus expected to lead to a destabilization of RGα and impaired complex formation, demonstrating that already a mild structural destabilization of RalGAP can be disease-causing.

## Discussion

The structural analysis of RalGAP reveals a unique tetrameric complex architecture. RGα2 and RGβ form a tail-to-tail heterodimer via their SD domains, and two heterodimers interact via head-to-head dimerization at the N-terminal domain of RGβ. The complex contains two active sites at the GAP domains of RGα2. The RGα2 homolog RGα1 is predicted to interact with RGβ in a very similar way and likely forms a complex with conserved structural characteristics. This is consistent with the comparable in vitro GAP activity of RalGAP iso-complexes containing either RGα1 or RGα2[11]. However, both RGα isoforms do not act in a fully redundant manner in vivo[32,35,39], which could result from differences in expression patterns or regulation.

RalGAP remarkably differs in its structure from the TSC complex, although the subunits RGα2, RGβ, and TSC2 show a conserved domain architecture and likely are derived from a common ancestral gene. In the TSC complex, TSC2 forms homodimers via the SD domains and binds the regulatory TSC1 subunit. TSC1 shows no similarity to RGα2 or RGβ and interacts with a long dimeric coiled-coil domain alongside the TSC2 homodimer and thus introduces asymmetry. TSC1 is not required to promote the activity of the TSC2 GAP domain per se as a minimal TSC2 construct was reported that has activity on Rheb comparable to the full-length TSC complex in vitro[45]. Instead, TSC1 is mainly responsible for the proper localization and function of the TSC complex in cells, by binding phosphoinositides, WIPI proteins, and TBC1D7[17,46-48].

Our study elucidates the relevance of RGβ for RalGAP function, identifying two relevant molecular mechanisms. First, the hug domain of RGβ binds RGα2 and thereby enables the GAP domain of RGα2 to adopt its active conformation. Presumably, RGα2 does not fold properly in the absence of β-hug, suggesting that the RalGAP subunits need to undergo a mandatory swap of their hug domains to complete their structure. Second, RGβ is required for the formation of the RalGAP tetramer, which does not influence catalytic activity in vitro but

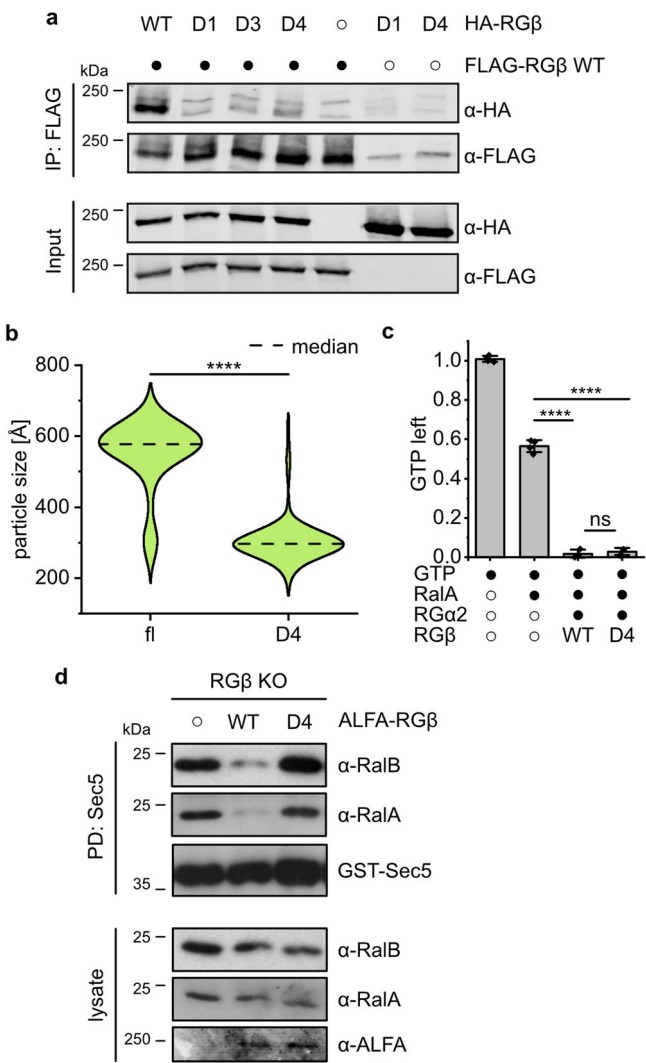

**Fig. 5 | Tetramerization is essential for RalGAP function. a** Co-immunoprecipitations of differently tagged RGβ full-length (RGβ^fl) and RGβ variants from transiently transfected HEK293FT cells. WT wild-type; D1: W65R; D3: V29E, V33Q, V37E; D4: V29E, V33Q, V37, W65R. **b** Particle length distribution measured by negative stain EM analysis of purified RGα2/RGβ^WT (n = 134) and RGα2/RGβ^D4 (n = 144) complexes. Statistical analysis: two-tailed Mann-Whitney test, ****$p < 10^{-14}$. **c** In vitro GAP assay of RalA GTP hydrolysis by RGα2 with RGβ^WT (n = 3 independent experiments) or RGβ^D4 (n = 3 independent experiments). Statistical analysis: two tailed unpaired t-test with Welch-correction, ****$p = 0.000046$ (no RG-RGα2/RGβ^WT); ****$p = 0.000092$ (no RG-RGα2/RGβ^D4); ns $p = 0.5127$ (RGα2/RGβ^WT-RGα2/RGβ^D4). GTP control n = 4 independent experiments; RalA control n = 3 independent experiments. Data are represented as means ± standard deviation. **d** Sec5 pull-down analysis of RGβ knock-out MEFs reconstituted with RGβ^WT or RGβ^D4.

functionality in cells. At this stage, we can only speculate why tetramerization is important, but it is likely a prerequisite for proper spatiotemporal control. This highlights how accurately RalGAP needs to be regulated in a physiological setting.

A previous study that mapped interaction sites within the RalGAP complexes identified a short peptide sequence in RGβ ('β-blockatide', residues E1125-T1162 in the RGβ sequence used in this study) that was able to bind RGα1 and reduce RalGAP complex formation and thereby protein stability[49]. We find that in the structure of RalGAP this peptide is part of the SD domain of RGβ but does not directly contact RGα2 (Supplementary Fig. 9). The effects observed with β-blockatide are thus likely caused by interference with RGβ folding rather than direct inhibition of subunit interactions.

Along these lines, it is interesting to note that many of the variants of uncertain clinical significance (VUS) in the *RALGAP* genes reported in cancer patients are likely to negatively affect complex function based on our structural assessment. The sheer number of variants makes a systematic experimental assessment hardly feasible. On the other hand, the automated computational categorization of variants by AlphaMissenese is currently not sufficiently reliable as it does not account for protein structure and function in the context of the assembled protein complex. The individual analysis of variants based on the experimental structure of RalGAP, we report here, thus provides a valuable basis for an initial assessment of VUSs.

Interestingly, no obvious mutation clusters can be identified but variants are spread over the entire lengths of all three *RALGAP* genes (Supplementary Fig. 8b–d). However, this does not rule out pathogenicity. The distribution of variants is reminiscent of mutations in *TSC* genes that cause TSC[50–52]. Pathological variants are also found throughout both *TSC1* and *TSC2* without apparent hot spot mutations. It is also interesting to note that just like in TSC, where mutations occur in the GAP complex but rarely in the cognate GTPase Rheb, no oncogenic Ral mutations are reported. Potentially, also Ralopathies may rather be caused by loss-of-function of the corresponding GAP protein.

Loss or reduction of RalGAP subunit expression has been shown to promote tumorigenesis in human cancer cell lines and mouse models and has been described to occur in a subset of pancreatic cancer patients[1]. A general function of RalGAP in cancer and the impact of RalGAP mutations has so far not been addressed. We report that a significant number of patient variants is expected to destabilize the complex or map to the functional important region that we newly identified, like the β-hug domain or the RGβ homodimerization site. Thus, our structure shows that many of the reported mutations will inactivate RalGAP and could abrogate its tumor suppressor function. It seems worthwhile to investigate (mutated) RalGAP as a tumor suppressor relevant for different human cancers more closely. However, a functional characterization of variants and the analysis of patient samples will be necessary to determine if the reported RalGAP variants are drivers or passengers in tumorigenesis. Our study provides the necessary structural framework for a systematic variant analysis that ultimately leads towards a better understanding of the relevance of RalGAP in cancer biology.

## Method
### RalGAP protein purification
DNA sequences encoding for RalGAPα2 and RalGAPβ were cloned into a p3xFLAG-CVM7.1 and pKH3 vector, respectively. κB-Ras2 DNA sequence was cloned into pCTAPa vector[53]. RalGAP complexes were expressed in Expi293F cells. For a 60 ml culture, a total amount of 60 μg plasmid DNA encoding the respective RalGAP subunits were mixed with 1500 μL PBS and incubated with PEI (1:3 DNA:PEI) for 15 min at room temperature before adding to 150 × 10^6 cells in a 250 mL flask. After incubating cells for 4 h at 37 °C and 8% $CO_2$, 3.5 μM valproic acid was added and the culture filled up to a final volume of 60 mL with Expi293 medium (Thermo Fisher Scientific). The cells were harvested after 3–4 days, and the pellets were snap-frozen in liquid nitrogen.

Cell pellets were thawed on ice and resuspended in 12 mL lysis buffer F (for full-length constructs: 50 mM Tris-HCl pH 7.5 150 mM NaCl, 1 mM $MgCl_2$, 5% Glycerol, 1% Triton-X100) or lysis buffer C (for minimal GAP constructs: 50 mM Tris-HCl pH 7.5 150 mM NaCl, 4 mM $MgCl_2$, 5% Glycerol, 0.3% CHAPS) supplemented with 1:50 protease inhibitor mix HP (Serva) and 0.5 μM DTT. Cells were incubated on a rotating wheel at 5 rpm, 4 °C for 20 min and debris was removed by centrifugation (12,000 × g, 45 min; 4 °C) and filtration (0.45 μM; cellulose acetate syringe filter). Clear lysate was passed four times over M2 α-flag beads (Sigma-Aldirch) equilibrated in lysis buffer. Beads were washed twice with 2 ml lysis buffer and twice with 2 ml RGH buffer (20 mM HEPES pH 7.5, 150 mM NaCl, 2 mM $MgCl_2$) and subsequently

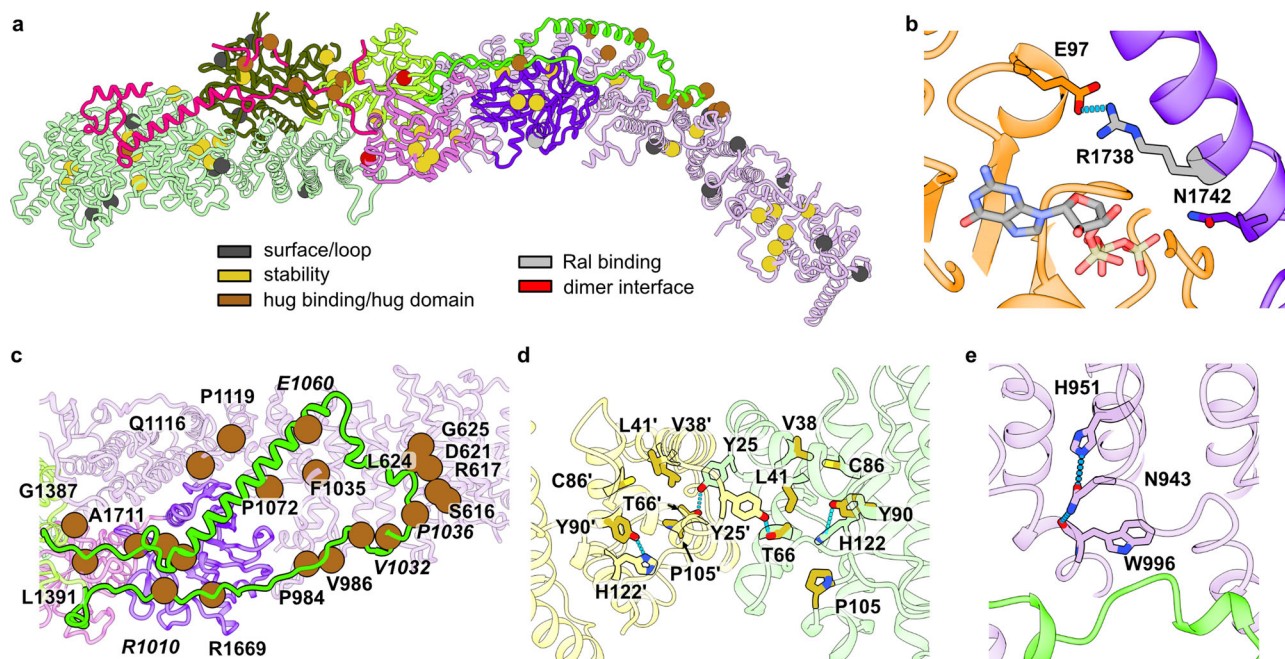

**Fig. 6 | Structure-based assessment of RALGAP variants. a** Mapping and structure-based classification of *RALGAPA2* and *RALGAPB* variants reported more than once in uterine and skin cancer patients. **b** Interaction of RGα2 catalytic helix residue R1738 with E97 of RalA. **c** Close-up of the β-hug/RGα2 binding site with the positions of patient variants shown as spheres. **d** Representation of the RGβ N-terminal homodimerization interface with the amino acids at positions of patient variants shown as sticks. **e** Stabilization of the RGβ^hug binding site of RGα2 by N943.

eluted with 0.1 μg/μl 3xFLAG peptide (ApexBio A6001 or TargetMol TP1274) in RGH and 1 mM TCEP were added. The combined elution fractions were concentrated with an Amicon Ultra Centrifugal filter (Millipore). For use in GAP assays, buffer was exchanged to RGA buffer (30 mM HEPES pH 7.5, 150 mM NaCl). For preparation of cryo-EM samples, the concentrated protein was loaded on a Superose 6 increase 5/150 column (Cytiva) equilibrated with RalGAP-EM buffer (20 mM HEPES pH 7.5, 150 mM NaCl, 2 mM MgCl₂, 1 mM TCEP). The peak fractions were directly used for grid preparation.

Murine RalA G-domain (aa 9-183) was amplified via PCR and inserted into pCDF6P vector via BamHI/NotI restriction sites. *Escherichia coli* BL21 cells were transformed with this construct and grown in TB media to an OD$_{600nm}$ of 0.9-1.0. Expression of the GST-tagged protein was induced by adding IPTG (0.5 mM) and cells were grown overnight at 16 °C. Cell pellets from a 2 l culture were resuspended in buffer L (50 mM NaH₂PO₄ pH 8.0, 300 mM NaCl, 1 mM MgCl₂, 1 mM DTT, 5% glycerol, supplemented with lysozyme, DNaseI, protease inhibitor mix HP (Serva)) and lysed (Microfluidizer). The lysate was cleared by centrifugation (39,000 × g, 45 min, 4 °C) and loaded onto equilibrated GSH-Agarose beads (Serva) three times and washed three times with 20 ml buffer L. The protein was cleaved from the beads with PreScission protease overnight and loaded onto a Superdex 75 pg 16/600 HiLoad size exclusion column (GE) equilibrated with buffer R (30 mM HEPES pH 7.5, 150 mM NaCl, 10% glycerol). Peak fractions were pooled, concentrated to 800 μM and stored at −70 °C.

## Negative stain grid preparation
Grids were glow discharged and incubated with 4 μl of protein solution for 2 min. Excess of protein solution was blotted away (Whatman paper No 1) and the grid washed twice with 10 μl water, once with 10 μL uranyl acetate (UA) and incubated with UA for 45 s before staining solution was blotted away. Grids were analyzed using a Talos L120C G2 TEM operating at an acceleration voltage of 120 kV at a magnification of 120k.

## Sample vitrification and cryo-EM data acquisition
For preparing the grids for cryo-EM, 4 μl sample at a concentration of 0.7 mg/ml were applied onto a freshly glow-discharged QuantiFoil 2/1 Cu300. The solution was incubated for 2 min, then manually blotted to remove excess liquid, and another 4 μl of protein solution was added, together with 1 μl 0.018% Triton-X100 solution. The excess liquid was then immediately blotted followed by vitrification in liquid ethane using a Vitrobot II automatic plunge-freezer (Thermo Fisher Scientific).

## Data acquisition
Datasets were acquired with a 300 kV Titan Krios G4 microscope (Thermo Fisher Scientific) equipped with an E-CFEG, a Selectris X energy filter and a Falcon 4i direct electron detector operated by the software EPU (Thermo Fisher Scientific). A total of 64,606 micrographs in five sub-datasets at 0°, 30°, and 42° stage tilt angles were collected in Electron Event Representation mode (EER) at a nominal magnification of 215k, corresponding to a pixel size of 0.58 Å/px. Most micrographs were collected at defocus of −0.5 to −4.2 μm. The Selectris X energy filter was used for zero-loss filtration with an energy width of 10 eV. A total dose of ~60 e⁻/Å² was aimed for by adjusting the exposure time of each sub-dataset to an appropriate value (between 2.7 and 3.6 s per micrograph). The details of dataset collection are summarized in Supplementary Table 1.

## Image processing and 3D reconstruction
EER movies were motion corrected in Relion5[54] using its own Motioncorr2[55]-like algorithm. CTF estimation was performed using CTFFIND4[56], and particles were selected using crYOLO[57]. The initial particle pick using the generalized model of crYOLO was suboptimal due to the elongated and partially overlapping particles. To improve the picking model and thereby achieve a reasonable selection of particles, Relion5 was used to perform multiple rounds of 2D classification, initial model generation, 3D-refinement, and re-extraction with improved centering, followed by retraining and repicking in crYOLO using the improved picking models.

With the thus optimized picking model, 5,519,589 particles were extracted from the motion-corrected micrographs by crYOLO, with a window size of 720 × 720 pixels, and binned to a box size of 360 × 360 pixels (1.16 Å/pixel). This binning factor was kept throughout further refinement. A subset of 887,720 particles were selected by 2D classification, keeping only classes that showed high-resolution features and complete particles which most likely were picked at somewhat similar positions along the particle. The latter was important as the elongated and relatively featureless mustache-like complex allowed origin offsets of up to several hundred pixels, thereby complicating subsequent 3D refinements. 3D classification into four classes resulted in only two approximately equally populated classes and two classes with only 37 and 252 particles, respectively. The better-resolved 3D class containing 420,975 particles was selected for further processing. 3D-refinement reached 4.1 Å resolution and could be improved to 3.98 Å after CTF and aberration refinement and another round of 3D-refinement in Relion5. Bayesian polishing and another round of CTF and aberration refinement, each followed by 3D auto-refine, further improved the resolution to 3.83 Å and a strongly improved map interpretability, which declined towards the box edges, as can be expected due to the elongated particle with dimensions exceeding the chosen box dimensions. We, therefore, performed a final round of multi-body refinement as implemented in Relion5, splitting the particles into three bodies, one in the center of the box, and two following the particle towards the box edges. This resulted in three partial maps with further improved map interpretability, and resolutions ranging from 3.79 to 4.35 Å.

## Model building

A model of the RGα2/RGβ complex was calculated using AlphaFold2[58] running on a local high performance computing cluster (PALMA II), docked into map from 3D reconstruction with ChimeraX[59] and energy optimized with ISOLDE[60]. In an iterative process, the model was optimized using real space refinement in Phenix Refine[60] against the composit map, and manual model building with Coot[61] using the full density map as well as the resampled multi body refinement maps. To generate a composite map of the three partial maps, these maps were resampled onto the initial consensus map in ChimeraX and combined by using the combine_focused_map tool from the Phenix suite.

AlphaFold3[40] was used to generate a model of RGα2 in complex with RalA, GTP, and $Mg^{2+}$. The model was superimposed with the experimental structure by aligning the GAP domains and the position of the Ral GTPase was combined accordingly with the experimental model. The energy of a composite model of the RalGAP core with Ral was minimized using the YASARA forcefield[62].

## Visualization and analysis of cryoEM maps and models

Structure visualization and analysis was done with ChimeraX (UCSF)[59,63]. Local resolution gradients within a map were calculated with RELION5 and visualized with ChimeraX. 3D angular distribution plots were generated in Relion5. 2D histograms of the angular distribution were generated using angdist[64]. The 3D Fourier shell correlation of cryo-EM maps was calculated using the remote 3DFSC processing server[65].

## GTPase activity assay

0.3 μM GAP proteins were incubated with 10 μM mouse RalA G-domain (aa 9-183) in assay buffer (30 mM HEPES, 150 mM NaCl, pH 7.5) supplemented with 20 mM EDTA and 1 mM DTT. 5 mM $MgCl_2$ and 50 μM GTP were added to start the reaction. The sample was briefly mixed and aliquots were snap frozen in liquid nitrogen at given time points and samples stored at −70 °C. Nucleotide ratios were analyzed by HPLC. The assay samples were boiled for 5 min at 95 °C and precipitated protein was removed by centrifugation. Of the cleared supernatant, 10 μl were injected in an Agilent 1260 Infinity II HPLC system equipped with an autosampler and a DAD HS detector.

Analytes were separated on an AMAZE HA mixed phase column (3 × 50 mm, particle size 3 μm; Helix Chromatography) protected by a SecurityGuard Cartridge (Gemini C18 4 × 3.0 mm ID; Phenomenex). The separation of G-nucleotides was achieved by a stepwise double gradient with increasing the buffer (200 mM $KH_2PO_4$ pH 2.0, 30–80%) and acetonitrile (15–20%) concentrations. The UV traces at 254 nm were used to monitor nucleotide elution. UV traces were semi-automatically analyzed with OriginPro 2024 (OriginLab Corporation). Traces were baseline corrected, the peaks corresponding to GDP and GTP integrated and the portion of GTP calculated. The remaining GTP amount was normalized to GTP at $t = 0$ h. In case of sub-stoichiometric complexes, the RGβ/RGα2 ratio was calculated from Coomassie stained SDS-PAGE band intensities (determined with ImageJ 1.54i) and the remaining GTP value compensated by this factor. Individual biological repeats were calculated from three technical repeats.

## Cell lines

HEK293FT (RRID:CVCL_6911) cells were obtained from Thermo Fisher Scientific (R70007). Mouse embryonic fibroblasts (MEFs) were generated from a C57BL/6 mouse embryo at day E12.5 and SV40-immortalized[26]. CRISPR/Cas9 mediated knockout of RalGAPβ was performed as described below[66]. All cell lines were cultured in DMEM high glucose (Sigma-Aldrich, D6546) supplemented with 10% FBS (Biowest, S1810), 10 mM L-Glutamine (Sigma-Aldrich, G7513) and 50 U Penicillin-Streptomycin (Thermo Fisher Scientific, 15070063) at 37 °C with 5% $CO_2$.

## CRISPR/Cas9-based generation of RalGAPβ knockout MEFs

For CRISPR knockouts sgRNA sequences were chosen with the CRISPick tool[67,68] (https://portals.broadinstitut.org/gppx/crispick/public) and primers designed with BbsI compatible overhangs. Annealed Oligos were cloned into px458 or px459 plasmids (gifts from Feng Zhang; Addgene plasmids # 48138 and # 62988[66]). SV40-immortalized WT MEFs were simultaneously transfected with px458 and px459 plasmids containing sgRNAs targeting upstream (RGβsg1 agaagcagtagtggtagtgt) and downstream (RGβsg2: gctgctaactccagttgcag) of exon 2 of RalGAPβ using jetPRIME® DNA and siRNA transfection reagent (Polyplus, 101000046). Twenty-four hours after transfection GFP-positive cells were FACS-sorted and returned to culture to recover overnight. Cells were then selected for 36 h with 2 μg/ml puromycin. Single cell clones were obtained by limited dilution. Successful deletion of RGβ was tested using PCR screening (mRGβ screen 2 F tgaaagggaaatgtcggaaa, mRGβ screen 2 R tgagttcctgccttggtttt) and confirmed by RT-qPCR using primers targeting inside the deleted exon (mRGβRTEx2_F cagtggctggtagtgagagt, mRGβRTEx2_R gcaacaccaaagccataatcc) and immunoblot.

## Reconstitution of RalGAPβ knockout MEFs

N-terminally ALFA-tagged RGβ$^{fl}$, RGβ$^{D4}$, or RGβ$^{Δhug}$ were cloned into a modified pITR-TTP vector[46] via MluI and NotI restriction sites. RGβKO MEFs were transfected with pITR, pITR-ALFA-RGβ$^{fl}$, pITR-ALFA-RGβ$^{D4}$ or pITR-ALFA-RGβ$^{Δhug}$ and the transposase expressing pCMV-Trp plasmid (9:1 ratio) using jetPRIME DNA and siRNA transfection reagent (Polyplus, 101000046) for stable reconstitution. Cells were selected for 48 h with 2 μg/ml puromycin and maintained with 1.5 μg/ml puromycin afterwards. To confirm exogenous expression, proteins were isolated with Bäuerle lysis buffer (20 mM Tris pH 8, 350 mM NaCl, 20% glycerin, 1 mM $MgCl_2$, 0.5 mM EDTA, 0.1 mM EGTA, 1% NP-40 supplemented with 1 mM DTT, Protease and Phosphatase Inhibitor Cocktail (Thermo Fisher Scientific, A32965, A32957) and analyzed by immunoblot.

## Co-immunoprecipitation

HEK293FT cells were seeded to 10 cm petri dishes and grown to 70% confluency. The medium was exchanged to starvation medium and

prepared DNA incubated for 4 h at 37 °C 5% $CO_2$. Medium was changed to growth medium. The next day, the medium was removed and the cells washed 1–2 times with ice cold PBS. 1 ml CoIP buffer (40 mM HEPES, 120 mM NaCl, 10 mM $MgCl_2$, 0.3% CHAPS, pH 7.4) supplemented with PIC (1:100) was added to the cells and cells collected using a gum wiper. Cells were incubated for 20 min on a stirring wheel at 4 °C before debris was removed by centrifugation. The clear lysate was taken and incubated with 20 µl 3xflag beads slurry (1:1 in lysis buffer) for 2–4 h. Beads were pelleted, the supernatant removed, and the beads washed three times with CoIP buffer. Beads were resuspended in 1× SDS-LD and analyzed by western blotting.

## Ral effector pull down
HEK293FT cells were transiently transfected as described previously. Cells were lysed in 800 ml SLB (50 mM Tris-HCl, 100 mM NaCl, 4 mM MgCl2, 1% Triton-X100, pH 7.5 at 4 °C). The Ral-binding domain of rSec5 was purified from *E. coli* as GST-fusion construct and loaded onto GSH beads. 20 µg GST-rSec5 were added to the cleared cell lysate and incubated (45 min, 4 °C, 5 rpm). The supernatant was removed and the beads were washed with SLB three times before resuspending in 1× SDS loading dye.

## Immunoblots
Proteins separated by SDS-PAGE (sodium dodecyl sulfate polyacrylamide gel electrophoresis) were blotted on Immobilon-FL (Millipore, IPFL00010) membrane using a semidry system (Trans-Blot Turbo Transfer System, 1704150, Bio-Rad). For imaging on the Odyssey CLx Imaging System (LI-COR), membranes were blocked for 1 h in Blocking Buffer (0.1% casein (Sigma-Aldrich, E0789) in 0.2× PBS). Primary antibodies were diluted 1:1000 in 1:1 PBS:Blocking Buffer with 0.1% Tween and incubated with the membranes at 4 °C overnight. Secondary antibodies were applied 1:4000 for 1 h in 1:1 PBS:Blocking Buffer with 0.1% Tween and 0.01% SDS: IRDye 800CW donkey anti-mouse (LI-COR, 926-32212) or IRDye 680RD donkey anti-rabbit (LI-COR, 926-68073). For chemiluminescent imaging, membranes were blocked for 1 h in 3% BSA (Serva Electrophoresis, 11930) in TBS-T (1% Tween). Primary antibodies were diluted 1:1000 in 1.5% BSA/TBS-T and incubated with the membranes at 4 °C overnight. Secondary antibodies were applied 1:4000 for 1 h in 1% dry milk powder (Carl Roth, T145.2) in TBS-T. Secondary antibodies used were Horseradish Peroxidase goat anti-mouse (Jackson ImmunoResearch, 115-035-044) or Horseradish Peroxidase goat anti-rabbit (Jackson ImmunoResearch, 111-035-045). For imaging on the ChemoSTAR Touch (intas) system, membranes were blocked for 1 h in 5% non-fat dried milk (Applichem) in 1× TBS. Primary antibodies were diluted 1:7000 (anti-flag), 1:2000 (anti-HA) and 1:1000 (anti-RalA) in TBS and incubated with the membranes at 4 °C overnight. Membranes were washed three times with TBS/T before HRP-coupled secondary antibodies were applied 1:25,000 in TBS and incubated for 2 h at RT (Polyclonal rabbit anti-mouse, Dako P0260; polyclonal swine anti-rabbit, Dako P0217). Membranes were washed and chemiluminescence induced with the SuperSignal West Pico PLUS detection reagent kit (Thermo Scientific, Ref 34577).

The following primary antibodies were used: anti-RalA (Proteintech, 13629-1-AP), anti-RalB (OTI2C4, Origene, TA505880), anti-RGβ (Proteintech, 28330-1-AP) anti-ALFA (Nanotag, N1582), anti-FLAG (clone M2, Sigma-Aldrich, F1804), anti-HA (clone C29F4, Cell Signaling Technology, 3742), anti-HA (clone 16B12, Biolegend) and anti-κB-Ras (provided by S. Ghosh, Columbia University[26]).

## Statistical analysis and reproducibility
Data curation and statistical analysis were done with OriginPro 2024 (OriginLab Corporation). Normality was tested with a Shapiro–Wilk test. Significance was tested with two-sample $t$-test with Welch-correction (non-equal variance assumed) for normally distributed samples and the Mann–Whitney test for non-normally distributed samples. Degrees of freedom values are given in the source data file. Results from representative examples were repeated at least three times independently.

## Reporting summary
Further information on research design is available in the Nature Portfolio Reporting Summary linked to this article.

## Data availability
The cryo-EM maps have been deposited in the Electron Microscopy Data Bank (EMDB) under accession codes EMD-53422 (composite map), EMD-53418 (consensus map), EMD-53419 (body 1), EMD-53420 (body 2), and EMD-53421 (body 3), respectively The atomic coordinates have been deposited in the Protein Data Bank (PDB) under accession code 9QWP (model). The source data underlying Figs. 3a–e, 5a–c, and Supplementary Fig. 6a, b are provided as a Source Data file. The coordinate of previously published structures of the TSC complex, the isolated TSC2 GAP domain and the Rap/RapGAP complex were retrieved from the PDB entries 7DL2, 6SSH, and 3BRW, respectively. Source data are provided with this paper.

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

## Acknowledgements

This work was supported by grants from Deutsche Forschungsgemeinschaft (DFG, German Research Foundation) to DK (KU2531/2, KU2531/6), CG (CRC1430, 424228829) and AO (OE531/4-1). The cryo-EM data were collected at "Cryo-EM SoN", the cryo-EM infrastructure of the University of Münster, funded by the DFG (project number 496113311). The cryo-EM data processing and AlphaFold2 modeling was carried out on the Palma II HPC (DFG INST 211/667-1) of the University of Münster. We thank Mark Nellist for the helpful discussions.

## Author contributions

D.K. designed the project. A.O., C.G. and D.K. supervised the study. R.R. purified proteins and prepared EM samples with M.C. and B.U.K.; B.U.K. collected and processed cryo-EM data with R.R.; R.R. performed structural modeling. R.R., L.H.A. and E.M. performed functional assays. R.R., A.O., C.G. and D.K. wrote the manuscript. All authors discussed the results, commented on the draft, and approved the final version of the manuscript.

## Funding

## Competing interests

The authors declare no competing interests.
