## [Transparent Peer Review file · Nature Communications]

Structure and mechanism of the RalGAP tumor suppressor complex

Corresponding Author: Dr Daniel Kuemmel

Version 0:

Reviewer comments:

Reviewer #1

(Remarks to the Author)

The enclosed manuscript submitted by Rasche and colleagues describes a structural and biochemical study of the RalGAP complex. Using cryo-EM, they have resolved novel quaternary structure that had not previously been foreseen within the field, which is of great mechanistic relevance, and therefore of significant interest for related protein complexes. The study is competently executed, if not at the highest resolution, and provides both a substantial advance and considerable numbers of novel mechanistic findings, which are well-placed within the context of the field in the discussion provided.

The results presented merit publication in Nature Communications, and I have no major concerns with any of the findings presented in the manuscript.

I enclose minor comments and issues below:

- Rather than simply moving directly to the half-complex, it would be appropriate to provide a low resolution structure of the complete complex from Fourier binned data (to overcome the processing issues mentioned - line 100), perhaps from negatively stained images if necessary, to allow the reader to understand the degree of the issues due to flexibility (line 98) mentioned in the results section, and compare this with the model in Figure 2F.
- Several sections discussing the modelled RalA structure (beginning line 154) should be clarified to make it clear whether the authors refer to experimental structure or the model in question, as parts of the paragraph are experimentally supported, while others are not.
- Figure 2G is not as clear as one might ideally prefer for the GAP and SD regions; might I suggest that the topology is clarified in the schematic through thickened connections between the same protein?
- It is not clear that the D4 mutant is an entirely well-behaved complex (SF7D). While I am in general agreement with the authors' statement on line 259, it might be appropriate to be as clear as possible about this in the main text, as there are other possible reasons for the observed results.

Reviewer #2

(Remarks to the Author)

RalGAP is a critical regulator of Ral GTPases and thus implicated in various cellular processes including vesicle trafficking, cell proliferation, and cytoskeletal organization. In this study, the authors presented the cryo-EM structure of the RalGAP complex. With proper biochemical analysis, they dissected the composition of the RalGAP complex and the importance of the interactions between the subunits in a cellular context. They also modeled the RalGAP-Ral-GTP complex by AF3, and analyzed the possible structural and biochemical consequences of the tumor sample-derived mutations on the subunits. Overall, the authors have made a breakthrough in solving the structure of an important GAP complex, which will be valuable in understanding its molecular function, regulatory mechanisms, and potential as a therapeutic target. This study warrants publication in Nature Communications, and I have a few comments mainly on the presentation of the data.

Supp Fig 1C

The two types of particles ('extraordinarily long' and 'narrow mustache-shaped') should be specified in the micrograph.

Line 102-104

'This yielded a map with overall 3.8 Å resolution and 4.4 Å for the focused maps of the hinge region, 3.8 Å for the central region, and 3.9 Å for the peripheral region of the complex'. I can more or less understand this sentence, but it is indeed quite blur. Please rewrite.

Figure 1

Besides the heterodimeric structure, the authors may consider to also show the two subunits R α 2 and R β separately. I strongly recommend the author to improve the color scheme. In current manuscript, the domains of the complex are all in cold colors and somewhat difficult to distinguish.

Line 153

This section focuses on a AF3 prediction of the RalGAP-Ral-GTP, which is not experimental data. I don't see the necessity of putting this content here and suggest the authors to mention it later, say right prior to the current last section about the disease-related mutations. In addition, the author could also make a prediction of the R α 1-R β version of the RalGAP complex based on their current complex structure.

Figure 2

As mentioned above, Figures 2A and 2B should appear later or be moved to the Supplement.

Line 199-201

How can the authors be sure that the AF3 model shown in Fig 2A is of 'high confidence'?

Line 218

'Show' should be 'showed'. Please check the grammar throughout the manuscript.

Figure 3

The quality of the blots shown in panel D should be improved. Also, I suggest the authors exchange the positions of the contents for Figure 2 and Figure 3, so that the biochemical data regarding the hug domain appears right after the structural description of it, and so for the tetramerization.

Line 270

I appreciate the efforts of the authors here for the delicate structural analysis of the mutants, but would like to remind them that the sparse mutations derived from the sequencing results of tumor samples are generally lacking the causal effect, unless they are found also in normal tissues of the same patient. Unlike those mutations on certain proteins found relative to inherited disease like neuromuscular disorders, sparse mutations from cancer patients can be just a random consequence of the instable genome of tumor cells that are already transformed for other reasons. It is not surprising as the authors described in the Discussion section that 'no obvious mutation clusters can be identified but variants are spread over the entire lengths of all three RALGAP genes, because any mutation that disrupts the folding of the subunits or composition of the complex would cause a loss of function. The authors should discuss more specifically how their structural data contribute to the pathological aspect of the RalGAP complex.

Figure 5

The mutations on the structural illustration in Individual panels are too small to see. Please enlarge the structural illustration and show only the spherical C α atoms.

Structural illustrations in Supplementary Figures should also be enlarged.

The author may include R α 1 into the Discussion section.

Version 1:

Reviewer comments:

Reviewer #1

(Remarks to the Author)

The authors have resolved all the issues I consider important, and I am happy to suggest the manuscript proceed to publication.

Reviewer #2

(Remarks to the Author)

The authors have fully addressed my concerns and I have no further comments, except that I feel Supplementary Figure 9 can be enlarged.

RESPONSE TO REVIEWERS

We thank the reviewers for their time and effort to evaluate our manuscript and appreciate the positive assessment and constructive feedback. The paper was carefully revised, and we hope that all concerns are addressed satisfactorily. In addition to the documents required for submission, we have included a combined pdf of the main part and supplement with all changes beyond minor edit highlighted to facilitate the re-reviewing process.

Reviewer #1 (Remarks to the Author):

The enclosed manuscript submitted by Rasche and colleagues describes a structural and biochemical study of the RaIGAP complex. Using cryo-EM, they have resolved novel quaternary structure that had not previously been foreseen within the field, which is of great mechanistic relevance, and therefore of significant interest for related protein complexes. The study is competently executed, if not at the highest resolution, and provides both a substantial advance and considerable numbers of novel mechanistic findings, which are well-placed within the context of the field in the discussion provided.

The results presented merit publication in Nature Communications, and I have no major concerns with any of the findings presented in the manuscript.

I enclose minor comments and issues below:

- Rather than simply moving directly to the half-complex, it would be appropriate to provide a low resolution structure of the complete complex from Fourier binned data (to overcome the processing issues mentioned - line 100), perhaps from negatively stained images if necessary, to allow the reader to understand the degree of the issues due to flexibility (line 98) mentioned in the results section, and compare this with the model in Figure 2F.

We fully agree with the reviewer and extensively tried to obtain a reconstruction of full RaIGAP, at least a low resolution. However, the intrinsic flexibility of the particle yielded 2D classes that could not be used for a 3D reconstruction in RELION. We took up the reviewer's suggestion and used Fourier binned data to attempt a classification and reconstruction with CRYOSPARC, unfortunately without better success.

We thus decided to include a negative stain image in Fig 1b to give a better impression of the full particle before focusing on the half-mustache. Furthermore, we included representative 2D class averages of full RaIGAP in the supplement and compared them to the class averages from the final reconstruction (Suppl Fig 1d) to give an impression of the flexibility of RaIGAP tetramers.

- Several sections discussing the modelled RalA structure (beginning line 154) should be clarified to make it clear whether the authors refer to experimental structure or the model in question, as parts of the paragraph are experimentally supported, while others are not.

Thank you for pointing this out. We have modified the paragraph (now l. 159-174) to clearly differentiate between predicted models and experimental structures.

- Figure 2G is not as clear as one might ideally prefer for the GAP and SD regions; might I suggest that the topology is clarified in the schematic through thickened connections between the same protein?

We have changed the Figure (now Fig 4d) accordingly.

- It is not clear that the D4 mutant is an entirely well-behaved complex (SF7D). While I am in general agreement with the authors' statement on line 259, it might be appropriate to be as clear as possible about this in the main text, as there are other possible reasons for the observed results.

Because the D4 mutant is active in vitro comparable to wild-type and as we are currently working with this recombinant protein complex in follow-up cryo-EM studies, we are relatively confident that the protein does not suffer from relevant structural problems. We of course cannot exclude secondary effects of the mutations in vivo and have thus included a disclaimer in the manuscript (l. 239).

Reviewer #2 (Remarks to the Author):

RalGAP is a critical regulator of Ral GTPases and thus implicated in various cellular processes including vesicle trafficking, cell proliferation, and cytoskeletal organization. In this study, the authors presented the cryo-EM structure of the RalGAP complex. With proper biochemical analysis, they dissected the composition of the RalGAP complex and the importance of the interactions between the subunits in a cellular context. They also modeled the RalGAP-Ral-GTP complex by AF3, and analyzed the possible structural and biochemical consequences of the tumor sample-derived mutations on the subunits. Overall, the authors have made a breakthrough in solving the structure of an important GAP complex, which will be valuable in understanding its molecular function, regulatory mechanisms, and potential as a therapeutic target. This study warrants publication in Nature Communications, and I have a few comments mainly on the presentation of the data.

Supp Fig 1C

The two types of particles ('extraordinarily long' and 'narrow mustache-shaped') should be specified in the micrograph.

We apologize, the description of the particle was misleading. There is only one kind of particle, and we now state: "The sample shows a homogenous set of characteristic extraordinarily long particles that resemble the shape of a mustache." (l 98-100)

Line 102-104

'This yielded a map with overall 3.8 Å resolution and 4.4 Å for the focused maps of the hinge region, 3.8 Å for the central region, and 3.9 Å for the peripheral region of the complex'. I can more or less understand this sentence, but it is indeed quite blur. Please rewrite.

We rewrote the paragraph (l 107-111).

Figure 1

Besides the heterodimeric structure, the authors may consider to also show the two subunits R α 2 and R β separately.

The structures of the RG subunits separately were included in Suppl Fig 4a,b.

I strongly recommend the author to improve the color scheme. In current manuscript, the domains of the complex are all in cold colors and somewhat difficult to distinguish.

Thank you for pointing this out. We changed the color scheme to achieve better contrast between interacting domains and to make the figures clearer.

Line 153

This section focuses on a AF3 prediction of the RalGAP-Ral-GTP, which is not experimental data. I don't see the necessity of putting this content here and suggest the authors to mention it later, say right prior to the current last section about the disease-related mutations.

As suggested, we removed Ral from the model of the full RalGAP complex. The modelling of Ral binding is now combined with the description of the (pseudo) GAP domains (new Fig 2, l. 159-174). Because these two aspects are closely related, we feel that this is the most appropriate position in the manuscript. Furthermore, before discussing the function of the b-hug domain, we believe that it is important to establish that it does not interact with Ral.

In addition, the author could also make a prediction of the R α 1-R β version of the RalGAP complex based on their current complex structure.

Thank you for the suggestion, a modelling of the R α 1/R β complex is now included before the description of RalGAP variants to corroborate the homologous structural roles of R α isoforms in the complex (new Suppl Fig 7).

Figure 2

As mentioned above, Figures 2A and 2B should appear later or be moved to the Supplement.

Because of its relevance for the investigation of b-hug function and the analysis of VUSs, we decided to keep the modelling of Ral binding, with modifications, in the main manuscript and hope that the reviewer agrees with our reasoning.

Line 199-201

How can the authors be sure that the AF3 model shown in Fig 2A is of 'high confidence'? The pLDDT and the positional error plot for the AlphaFold3 model are

now included in Suppl Fig 5 (panels a,c). These metrics show that both the folding of the individual domains and the positioning of Ral relative to the Rga2 GAP domain have high confidence scores. The model can thus be considered reliable within the limitations of structure prediction. The paragraph was also changed to better differentiate between predicted models and experimental structures (see also reviewer 1).

Line 218

'Show' should be 'showed'. Please check the grammar throughout the manuscript. Thank you, we have corrected the error and carefully proofread the revised manuscript.

Figure 3

The quality of the blots shown in panel D should be improved.

Fig 3D was replaced with blots of better quality from a different repeat of the experiment.

Also, I suggest the authors exchange the positions of the contents for Figure 2 and Figure 3, so that the biochemical data regarding the hug domain appears right after the structural description of it, and so for the tetramerization.

Thank you for the suggestion, the order of the figures (now Fig 3 and 4) was changed.

Line 270

I appreciate the efforts of the authors here for the delicate structural analysis of the mutants, but would like to remind them that the sparse mutations derived from the sequencing results of tumor samples are generally lacking the causal effect, unless they are found also in normal tissues of the same patient. Unlike those mutations on certain proteins found relative to inherited disease like neuromuscular disorders, sparse mutations from cancer patients can be just a random consequence of the instable genome of tumor cells that are already transformed for other reasons. It is not surprising as the authors described in the Discussion section that 'no obvious mutation clusters can be identified but variants are spread over the entire lengths of all three RALGAP genes, because any mutation that disrupts the folding of the subunits or composition of the complex would cause a loss of function. The authors should discuss more specifically how their structural data contribute to the pathological aspect of the RalGAP complex.

Thank you for this comment, and we fully agree. It was not our intention to suggest causality between the occurrence of RalGAP variants and the progression of cancer, and we regret if this impression arose.

The point we are trying to make is that it appears worthwhile undertaking such studies to investigate this question. Our structure shows that many of the reported

mutations will inactivate RalGAP and thus abrogate its tumor suppressor function. Also, the distribution of variants in RalGAP is reminiscent of mutations in the closely related TSC complex that are established to be disease causing. Of course, a functional characterization of variants and investigations with patient samples will be necessary to determine if the reported RalGAP variants are drivers or passengers in cancer, and we have initiated respective studies.

With the following changes to the manuscript, we hope to provide a more balanced view:

- the implication of relevance in tumor development was removed from the abstract
- The graph showing mutation frequency is moved to supplement (now Suppl Fig 8A) to avoid the impression of a direct link to cancer.
- The comparison to TSC complex mutations is removed from the results section to preclude drawing a direct analogy. TSC is included in the discussion to point out that the presence of mutation hot spots is not a prerequisite for disease-causing genes.
- We have toned down the last paragraph of the discussion and added a disclaimer to prevent any misconceptions about the implications from our structural analysis.
- Discussion of the importance of structural information to understand RalGAP (patho)biology was added.
- As a further example for the usefulness of the structural information, the characterization of the R α 2 variant R1738H that likely affects Ral binding was included (new Fig 6b).

Figure 5

The mutations on the structural illustration in Individual panels are too small to see. Please enlarge the structural illustration and show only the spherical C α atoms. Structural illustrations in Supplementary Figures should also be enlarged. We have enlarged and changed the images accordingly.

The author may include R α 1 into the Discussion section.

Thank you for the suggestion, the structural similarity of R α 1 and R α 2 is now discussed (l. 304-310)

RESPONSE TO REVIEWERS

We thank the reviewers for their time and effort to evaluate our manuscript and the positive assessment. The paper was revised to address the remaining suggestions.

Reviewer #1 (Remarks to the Author):

The authors have resolved all the issues I consider important, and I am happy to suggest the manuscript proceed to publication.

Reviewer #2 (Remarks to the Author):

The authors have fully addressed my concerns and I have no further comments, except that I feel Supplementary Figure 9 can be enlarged.

We have enlarged Supplementary Figure 9 as suggested.